# Why Does Israel Lead the World in COVID-19 Vaccinations? Applying Mass Casualty Event Principles to COVID-19 Vaccination Programs

**DOI:** 10.3390/ijerph18105362

**Published:** 2021-05-18

**Authors:** Carmit Rapaport, Isaac Ashkenazi

**Affiliations:** 1NIRED—Institute for Regulation of Emergency and Disaster, College of Law and Business, Ramat Gan 5110801, Israel; carmit.rapaport@gmail.com; 2Department of Geography and Environmental Studies, University of Haifa, Haifa 3498838, Israel; 3Faculty of Health Sciences, Ben-Gurion University of the Negev, Beersheba 8410501, Israel

**Keywords:** COVID-19, mass casualty events, crisis management, vaccination, Israel, population behavior

## Abstract

The article examines Israel’s experience in managing the COVID-19 vaccination program beginning in December 2020. Utilizing principles of mass casualty event management, such as triaging, flow of casualty care, and flexibility (among others), we analyze Israel’s vaccination program. The successful Israeli experience was found to be based on timely coordinated and adaptive health system logistics and operations, as well as cooperative population behaviors.

## 1. Introduction

Managing a pandemic crisis such as COVID-19 poses huge challenges to countries. Such challenges include: coping in uncertain settings given the unknown nature of the pandemic’s distribution and its potential health effects, the need to gain public adherence to movement and gathering restrictions in order to control and prevent outbreaks, and the immediate need to adjust the healthcare system’s capacities to mitigate overwhelm. Mitigation policies posed a “symmetrical solution” to an “asymmetrical problem” [1]. These have led to massive disruptions of social and economic systems, with an ongoing dispute on the “acceptable loss” [2], i.e., finding the fragile balance between preventing the healthcare system’s insufficiency while opening the economy and maintaining life routines as much as possible.

Given these challenges, it has become clear that managing the COVID-19 pandemic demands a multi-organizational response based on rapid decision making in a setting of uncertainty, risk management, coordination, and continuous flexible adaptation. A decisive factor in pandemic management is vaccinating the population [3]. An effective vaccine might allow for a quicker return to normalcy. Once the vaccine is developed and approved, administering it to the population is a complex process involving a multi-organizational collaboration, risk management, logistics, and continued adaptation to the changing conditions under time constraints and emotional pressure.

In this article, we analyze the recent experience of the State of Israel in providing COVID-19 vaccinations to the public and suggest that decision making during emergencies (such as the COVID-19 pandemic) should follow the mass casualty events methodology [4] including on-spot triage and decision making (pre-hospital care), logistics (transportation and distribution), and professional execution (hospital-based acute care). The operation of the vaccination program represents the process of making an irregular decision of vaccinating the entire population immediately after the vaccine is approved by national regulatory authorities (“what to do”), arranging logistics (“how to do it”), and maintaining continuous flexibility to adapt to unexpected issues that may arise during the actual execution of the operation.

The Israeli case of the COVID-19 mass vaccination management is an interesting case study for evaluating and examining crises management principles, especially in a global context. As of May 2021, 55% of the Israeli population (age 16 and up) was fully vaccinated (Israeli Ministry of Health, 2021) (https://datadashboard.health.gov.il/COVID-19/general/ (accessed on 12 May 2021)). Within the first two months of the vaccination rollout (January–February 2021), 36% of the entire population had received the vaccine (Israeli Ministry of Health, 2021). The vaccinations had a dramatic effect on decreasing the number of infections, the number of positive cases, and the number of severe positive cases. As such, what can be learned from the Israeli experience of the COVID-19 vaccination program?

## 2. Mass Casualty Event Management and Response

The Israeli case will be described based on the fundamental principles of mass casualty event (MCE) management [4].

According to Ashkenazi et al. (2010), MCE management deals with five major challenges:Leadership;Prehospital care (immediate responders and first responders);Patient transport and distribution to medical facilities;Hospital care;Community and media relations.

The model includes the sequence of MCE medical treatment: pre-hospital care, transportation to the hospital and hospital care, and adds two important management components: leadership and community/media relations. There is also a differentiation between the MCE management phases: before the event, in the preparedness stage, and during the event, when providing medical care in both the pre-hospital and in the hospital. This differentiation highlights the various actions that must be carried out in order to be properly capable in responding to the MCE, in terms of first responders’ skills, communications and collaborations with other units, equipment and logistics, procedures and protocols, security and mental coping. These capabilities are integral in developing personal leadership abilities and defining the chain of command in all organizations’ personnel who are included in the MCE response. At this stage, attention should also be paid to preparedness activities such as training and building collaborations with local stakeholders, such as community leaders, for example. Furthermore, at the same time, the healthcare system itself should be prepared for effective care in both the pre-hospital and hospital phases for patients with various needs. This includes preparing for a medical surge, managing capacities and capabilities, training medical personnel for triaging and medical care, and defining possibilities for transportation to the hospitals.

MCE medical management is based on effective coordination in three phases [4]: (1) actions made at the disaster scene; (2) patient distribution to medical facilities; and (3) definitive care at medical facilities. During these phases, the guiding principles are: situation assessment, quick decision making and triaging, and “fast, light and smart” responses.

In contrast to routine care for an individual, the main goal in an MCE is to save as many casualties as possible, and decrease the minimum future damage such as physical and mental disabilities. Therefore, the reaction should be based on rapid situation assessment and immediate decision making. During an MCE, there is a structured trade-off between the willingness to provide the best and the quickest care. When time and resources are limited, the “quickest” response is preferred over the “best”—as more patients could be saved. Furthermore, as time is the most critical factor, the response should be “fast, light and smart”- acting quickly in terms of deciding what to do and how to do it immediately, while also being “light” and “smart”—adapting to the changing conditions as quickly as possible (for example, utilizing on-the-spot resources, such as tourniquets, recruiting untrained bystanders to provide help, etc.). As Ashkenazi et al. (2010) [4] noted the standards of care might be altered in order to save the most lives. Timely arrival of a critical condition patient at the hospital makes the difference between life and death. Although ambulances and EMS are designated for this mission, in a mass casualty event, when there are not enough resources, even a taxi could transport a patient to the nearest hospital and save his/her life.

We suggest that, based on the Ashkenazi et al. (2010) [4] model, there are three basic components of MCE management in the context of the vaccination operation: (1) decision making (leadership), (2) mission execution (managing the event: pre-hospital, transportation to hospitals and in- hospital care), and (3) continuous flexible adaptability (community and media relations). Between these components, we detect two important intra- and inter-agencies characteristics of managing complex and uncertain situations: cooperation and coordination and flexibility (Figure 1).

## 3. Decision Making

In the vaccination case, as during an MCE, event managers are required to make rapid strategic, operative and tactical decisions regarding the desired outcomes under limited time, high risk, constrained resources, fluctuating public trust, harsh criticism of the media, etc. This requires continuous flexible adaptability to changing conditions on the go. Given the extreme time pressure and uncertainty, there is a need for meta-leadership [4,5]. This includes the emergence of the “meta-leader”, i.e., a person (or persons) who mentally and cognitively adjusts to the evolving situation and acts effectively when others might be in shock or in a state of paralysis. The meta-leader also develops a situational awareness and leads others (including bystanders who are present at the scene) while leading the mental processes which reflect the shared reality and sense-making of the situation, as agreed among the different stakeholders [6]. The meta-leader will lead both horizontally (i.e. will initiate and enhance collaboration between different formal and informal agencies), and vertically, according to the command chain. Eventually, the leadership in such an event demands cross-system connectivity, which reflects the flexible adaptability as it connects systems, persons, information sources and resources according to the needs of the current evolving situation. The decision making processes are reflected in the Israeli vaccination operation:

**Managing the urgency while anticipating the next levels**: At early stages of the pandemic, it became clear that an effective vaccine would be the ultimate solution for containing the pandemic and mitigating its distribution. Being highly contagious, the COVID-19 pandemic put certain populations at high risk. Movement restrictions and lockdowns can be effective for limited periods of time only, due to economic and social impacts on populations and pressures on decision makers. Sociologically, the fear of a “shared enemy” has increased the understanding that solidarity and collective behavior towards protection are important factors in mitigating the pandemic. An effective vaccine allows for a quicker return to normalcy, but only if a significant amount of the population gets vaccinated.

Israel quickly responded by contracting with the main companies developing the vaccines, and assuring a sufficient quantity from various companies to its entire population [7]. This was done before the U.S. Food and Drug Administration (FDA) approved the vaccines, and follows one of the basic principles of MCE management: during the emergency phase, it is vital to handle the most urgent issue while mitigating direct threats. However, at the same time, attention should also be given to “what comes next”—this means, how to preempt the factors that caused the threat and minimize the chances of them reoccurring. 

In Israel’s case, managing the pandemic on a daily basis (including containing outbreaks by applying stay-at-home orders and purchasing medical equipment for hospitals), occurred in parallel with negotiating and purchasing the vaccines, even though they were still being developed. Once the FDA approved the vaccine, and before it was supplied to Israel, the Ministry of Health began communicating with the public and called on citizens, via a massive campaign, to get vaccinated and receive a “green pass” which would allow entrance to public places [8]. The campaign also aimed at gaining the public’s trust in the vaccine while highlighting its safety and advantages. This reflects the “what comes next” approach: although there was not much information about the vaccines in general, the public had received the entire information available and when the vaccines arrived, it was only a matter of time until people surged to receive them. The campaign (which focused on the vaccine’s potential to allow a quicker return to normal life) and the ensuing public debate created a high demand for the vaccine, which was limited in quantity. Furthermore, the government would also take responsibility for the treatment of citizens with any potential side effects that might result from the vaccine.

**Prioritizing at-risk populations**: As in MCE management, the principle of triage (i.e., assessing and sorting casualties according to the severity of their injury, condition and resources in order to provide them with urgent and appropriate medical care), is based on ethical grounds. The triage aims at saving as many lives as possible, while prioritizing those in severe conditions over other patients. The Ministry of Health defined the procedure of prioritization of the vaccination groups and sub-groups based on MCEs ethics (Ministry of Health "The first ones to have the COVID-19 vaccine within the ‘Ten Katef’ Vaccine Campaign, 2020 (Hebrew). https://www.gov.il/en/departments/news/16122020-01/ (accessed on 12 May 2021)). 

In the case of Israel’s management of the vaccine program, the prioritized populations were, first, the elderly (citizens over the age of 60), nursing home residents, and healthcare personnel. The Ministry of Health also decided which healthcare organization is responsible for vaccinating each group [7]. At the same time, physically-challenged elderly who are grounded in their homes were not prioritized, as vaccinating them in their homes would require allocating resources that would minimize the capacity to implement the mass vaccination program to most of the elderly population. The basic triage principle applies here: first save those who need your immediate help in order to survive. Each prioritized group includes multiple sub-groups that also need to be prioritized. Eventually, the decision was based on prioritizing groups as well as the number of potential survivors.

**Centralized management of the healthcare system**: The Israeli healthcare system is based on four health maintenance organizations (HMO) which are directly subordinated to the Ministry of Health. These community-based, non-profit organizations operate healthcare clinics in every settlement in the country, providing all health services in the community, from routine inoculation of newborns and children to experts’ consultations and treatment. All citizens’ medical records (beginning from their birth days), are collected and assembled, which allows constant epidemiological tracking of the population. Furthermore, all healthcare systems units—hospitals, clinics, pharmacies, diagnostic centers and laboratories are connected to a unified database managed by the Ministry of Health, which allows connection between preventive medicine, medical care, and social and health services for each citizen. Health services are provided nationally for each citizen, free of charge, including vaccinations, by law. Further, the HMOs routinely carry out nation-wide prevention programs such as seasonal flu vaccines [7], obesity awareness, and active lifestyle campaigns [9]. Therefore, the decision to vaccinate the population was made quickly, given this centralized healthcare system which allowed an immediate full support of healthcare professionals who could apply existing mechanisms to distribute the vaccines to citizens.

**Past experience with mass vaccinations**: Israel has previous experience with mass vaccinations for polio, measles [10] and smallpox [11], as well as influenza vaccinations [12] each winter.

**The collaboration with vaccination companies**: For the pharmaceutical companies, collaborating with Israel sets an ideal testing field for the vaccinations: the national level databases with full medical records of the patients allows for excellent tracking of side effects as well as of vaccination effectiveness. Israel, in turn, was prioritized in the vaccination supplies that might promote its ability to overcome the pandemic and reopen the economy as soon as possible (https://www.npr.org/2021/01/31/960819083/vaccines-for-data-israels-pfizer-deal-drives-quick-rollout-and-privacy-worries/ (accessed on 12 May 2021)).

**Purchasing the vaccines**: Israel began negotiating with pharmaceutical companies at early stages of the pandemic [7]. As a result of a risk assessment, the government had purchased vaccines from all three main companies: Pfizer, Moderna, and AstraZeneca, well before the FDA's approval and even before the final trial results were published. Additionally, the Israeli National Biological Center also developed a vaccine. Once the decision to vaccine the entire population had been made, all resources were channeled to achieve this, including purchasing needles, syringes, and other equipment.

## 4. Mission Execution

**Storage and distribution**: Once the decision to purchase vaccine doses for the entire population had been made, the logistical procedures were carefully planned and exercised- from the moment the packages arrived in the country, to their storage, to their distribution to the vaccination areas around the country, and overall safety control. These procedures were prepared and simulated with all of the involved organizations to ensure the efficiency and speed of the mission. This “moving assembly line” technique allows for excellent application of the vaccination procedures, saves time and resources, and enables effective inoculation of hundreds of thousands of people per day.

**Vaccination area operation procedure**: As in MCE management, there is a need to create a “flow of casualty care”— a dynamic flow in which the patient enters at one point, receives the treatment (or being triaged in the case of MCE), and eventually receives the adequate solution to his/her condition. In the case of the vaccination program, the patient enters the vaccine area and goes through the flow of care: identification of the patient, confirmation that the patient belongs to the prioritized groups, inserting the patient’s details into the database, administering the vaccine, and directing the patient to wait in a waiting room for 20 min.

## 5. Continuous Flexible Adaptability

**Recruiting nurses to administer the vaccine**: Given the high demand and need to vaccinate the most citizens as quickly possible, there was an emergent need to recruit more nurses who could administer the vaccine at vaccination centers. This was done by a quick training of nurses, medics and paramedics for administering COVID-19 vaccines and placing them in vaccination centers [7]. The Israel Defense Forces also recruited military medics and paramedics who were trained to administer the vaccines [7]. The ability to increase the capacity of healthcare professionals while handling an MCE is an important factor in crisis management. According to Ashkenazi et al. (2010) [4], as the number of casualties exceeds the available resources during an MCE, there is a need to manage an expected surge of patients. To do so, the healthcare system should increase its capacity, i.e., its ability to manage the increased number of patients. In addition, the healthcare system should manage the patients’ surge capability, which means being able to provide adequate care to patients with diverse and infrequent medical needs. Given the high demand for vaccinations in Israel and the willingness to vaccinate all the risk populations as quickly as possible, one can say that this vaccination operation is parallel to a patient surge. Therefore, increasing the number of professionals who take part in administering the vaccinations allows for both improved capacities and capabilities to vaccinate substantial percentages of the population. In terms of the capabilities—allocating nurses and paramedic to provide vaccines in nursing homes and distant settlements—allows nondiscriminatory and efficient rollout of the vaccination operation.

**Maximal usage of the vaccine doses**: In each vaccination center, designated personnel prepare the vaccine doses for the nurses for maximal extraction of each dose, which makes this process more efficient in terms of time, logistics, and queue management. Moreover, at the end of each day, each vaccine center tried to avoid destroying leftover doses as much as possible; the HMOs therefore called on members of all ages to arrive to the vaccine centers and receive the “leftover” doses, regardless of the age prioritization. This demanded a rapid, flexible and adaptive response in terms of evaluating the amount of leftovers versus the number of daily vaccines, getting in touch with potential population groups, and maximizing the vaccinations administered per day. Furthermore, in cases where specific age groups were less responsive to receive the vaccine, the prioritized age was lowered immediately after this trend was detected. For example, after realizing that the 50–60 years old citizens were less willing to get the vaccine, the prioritized age was lowered almost immediately to the 40 years old group. All healthcare facilities in the country, including public and private hospitals, clinics, and mobile service units, provided the vaccines, first to the prioritized groups, with efficient leftover management enabling any daily surplus of doses to be offered to those who were interested. This is an example for managing the medical surge as in an MCE: the healthcare system expands its capacity (to meet the demand) and its capability (to treat all cases, be it complex patients, hospitalized patients, nursery inhabitants, among others).

**“Closure”**: As the full immunization is reached only after receiving two doses, it was essential to manage the registration and administration of the second dose, as well as registering the vaccine in the personal medical records of receiving citizens. After receiving the first dose, the citizen got an SMS indicating the date of the first vaccination, the vaccines details (the producing company and the place the vaccine was administered) and the follow-up appointment for the second dose. In addition, the SMS includes is a link to a form asking about the presentation of side effects. After the second dose, the citizen receives a vaccination certificate, which indicates the vaccination details, expiration date, and personal details (Figure 2).

## 6. Community and Media Relations

**Population cooperation**: The Israeli population is highly cooperative with vaccinations in general, and in children in particular [13]. The health awareness is high, and a significant percentage of the population is trained for administering basic first aid at schools and even for advanced medical aid during their mandatory army service. Furthermore, the Israeli population shows high levels of trust in the emergency systems—during routine [14] and during the COVID-19 pandemic [15]. A survey conducted in the midst of the pandemic in Israel (August 2020) revealed that the most trusted public entity is the Ministry of Health (53% answered that they trust the ministry while only 30% answered that they trust the government) (Israeli & Deitch, 2020 https://www.inss.org.il/publication/coronavirus-inss-survey/ (accessed on 12 May 2021). 

Further, Israelis were found to be compliant with the official protection instructions due to fear of being infected, fear of being fined, unwillingness to endanger high-risk populations, and general citizen compliance with the law [16]. However, in a survey conducted in November 2020, a month before the vaccination operation started, when asked whether they would be willing to get vaccinated at the early stage of the vaccination program, 21% answered that they were willing to get the vaccine with certainty. Another 19% answered that they thought that they might be willing to get vaccinated at early stages of the vaccination program, and 25% and 27% answered that they thought that they were not willing or declared that they would not get the vaccine at early stages, respectively [16]. These data show that Israelis separate between their attitudes towards politicians and governmental offices and the pandemic decision makers. This might be one possible explanation for the high demand for vaccinations.

As in MCEs, an important factor in a successful management of the emergency is the communication with the public. Such communication is critical due to its role as mediator between the authorities and the public, to provide ongoing and updated information about the current situation and potential upcoming situation/s, as well as what is expected from the citizens. In terms of the vaccines, surveys conducted at early stages of the pandemic outbreak in Israel (March 2020) have shown high levels of “vaccine hesitancy” [17], which was related to the vaccine safety and effectiveness, and perceiving the disease as not dangerous to one’s health. As in the cases of MCEs, such attitudes, fears, and increased uncertainty might have a negative effect on the public’s ability to cope with the evolving situation and on community resilience [4]. The Ministry of Health’s campaign was present in all media channels, including social media, and included daily reports of the number of those vaccinated [7]. Clear messages highlighted the importance of getting the vaccine, and information detailing the specific vaccine process was widely published (including where citizens should go and what they should do to get vaccinated). Additionally, the communication with the public was generated by one source- the Ministry of Health- which is parallel, in the case of MCEs, to the Hospital Incident Command System (HICS) [4]. Managing the media in a centralistic manner allowed the public to refer to official and updated information and knowledge.

Furthermore, an important incentive for vaccination is the vaccination certificate (“green pass”) which is given to each citizen after receiving the second dose of the vaccine. Legal and ethics experts, together with policy makers, are discussing the questions of whether and to what extent can those who carry this certificate participate in public activities, such as mass cultural and sporting events, tourism and recreation (given that the vaccination is voluntary).

Lastly, another cause for the public’s cooperation lies in the prevalent mentality of being a proactive responder and not a victim. This resilient coping mechanism is rooted in the Israeli experience with protracted terror campaigns and is expressed by increased levels of immediate helping behavior (helping bystanders and national humanitarian aid in disaster zones) [18].

## 7. Discussion—After Action Analysis

In this article, we described the Israeli management of the COVID-19 vaccine program using the framework of Mass Casualty Events. The vaccine operation in Israel has been effective, as reflected in the high immunization rates and the decreased pandemic metrics. As shown in the analysis, several components of MCE management were applied to the COVID-19 vaccine operations in Israel. As in MCEs, the time factor was critical and decision makers applied “fast, light and smart” actions. Decisions were made quickly, while prioritizing at-risk populations such as in medical triage. The healthcare system utilized all of its resources and created a comprehensive operation well before the vaccines had arrived in Israel, through the response phase (i.e., the logistics and administration of the vaccine to the public), while constantly adapting to the changing conditions to maximize the effectiveness of the entire program.

Another important aspect of MCE management is coping with an expected medical surge. Systematic preparedness for such a case includes: planning, continuous flexible adaptability in managing medical capacities and capabilities, redundant equipment, and high-quality triaging even while altering the standards of care. In the case of Israel’s vaccination program, the efforts focused on vaccinating as many citizens as possible with maximal efficiency of the process (vaccinating persons in nurseries and institutions, utilizing every dose, etc.). This demanded effective coordination between the healthcare systems’ partners and continuous flexible adaptability in irregular cases.

Finally, a central component of this successful experience with the COVID-19 vaccination is the technology adoption by both the decision makers and the citizens. Utilizing the vaccines- an innovative technology- in such high volumes, although new to the world, reflects an outstanding belief in technology in general, and in medical technology in particular. The COVID-19 pandemic sets an opportunity to remove bureaucratic obstacles and enhance innovation. Through early adoption of advanced technologies, a country can gain a significant competitive advantage, which in the case of COVID-19 pandemic can make the difference between countries who will regain their normalcy and those who still struggle with the virus.

In summary, the successful and intensive COVID-19 vaccination program experience in Israel can serve for evaluation of emergency management procedures as well as of MCE management procedures. As demonstrated with Israel’s vaccination program- a timely, flexible, adaptive and coordinated decision-making structure and execution strategy could very well enhance countries’ responses to emergency situations.

## Figures and Tables

**Figure 1 ijerph-18-05362-f001:**
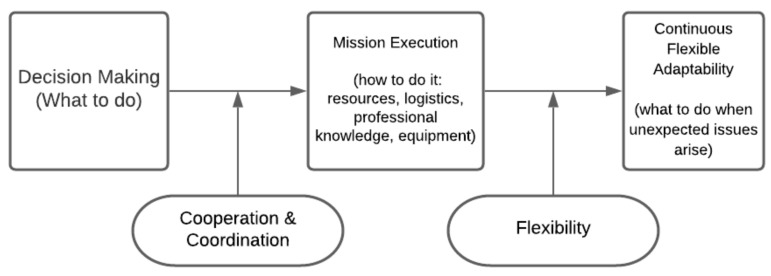
Components of emergency response during uncertainty.

**Figure 2 ijerph-18-05362-f002:**
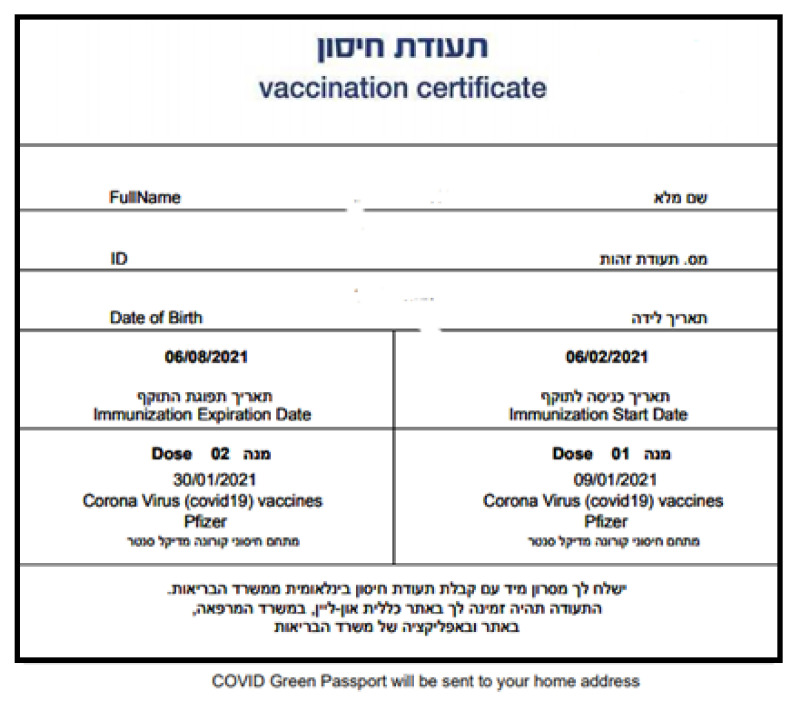
Example of a Vaccination Certificate.

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
