# Peer review of "Why Does Israel Lead the World in COVID-19 Vaccinations? Applying Mass Casualty Event Principles to COVID-19 Vaccination Programs"

_ijerph, 2021, doi:10.3390/ijerph18105362_

Round 1

Reviewer 1 Report

The article analyzes the recent experience of the state of Israel in its COVID-19 vaccination operation. The authors contend that decision making processes during emergencies, such as COVID-19, should follow mass casualty event methodology.

The article is both interesting and important. It assists to provide initial understanding of the managing principles that helped Israel to successfully implement its vaccination operation during an emergency caused by an unfamiliar epidemic and in an uncertain, unpredictable context. Nevertheless, the article needs further clarifications and editing work. In the following paragraphs I will try to elaborate on my suggestions for editing the article.  

  • Introduction – the introduction lacks a funnel-like structure. After introducing the challenges posed in managing a pandemic crisis it needs to describe and typify the vaccination operation in Israel and then present the article's contention.

Since the authors lean on a previous model of principles for mass casualty event (Ashkenazi et al., 2010), these principles should be presented clearly. Further then the authors should present the reasons this principal model needed translation and adaptations and specify their offered adapted principal model clearly – both in the text and through drawing a figure. The figure can take two versions – a "thin" one containing the main principles and a more elaborated one that will also present sub-principles of the model.   

Note that Ashkenazi et al. (2010) use the definition: mass casualty event in their article, and not incident, and specify 5 principles while the authors mention 4. Also, it is highly important to mention that decision making involves processes since it is through and upon these processes that decisions, strategies, practices etc. are being agreed upon and implemented.

The authors should provide and define what was the main reason(s) that stood behind the purpose of "opening the economy and keeping life routines" – to my opinion the purpose was to preserve Israel's functional continuity and to prevent further damage to the economy and wellbeing of Israelis which approximately 40% of them reported a mental deficiency of some sort caused by side effects of the pandemic (quarantine, social isolation, job losses etc.).  

Paragraphs (sub-titles) 2-4: For the paragraphs to be understandable to readers I strongly suggest presenting the central principle at the beginning of each paragraph and then exemplify it through using decision, practices etc. implemented in the Israel vaccination operation and through using concrete examples.

In paragraph 2 pls. specify why, from a sociological point of view and in your opinion, an effective vaccine was the ultimate solution for containing the pandemic in Israel. I believe that a social explanation for this conclusion stands in affinity to other social factors influencing diverse steps, practices and procedures chosen by Israeli decision makers during the pandemic in Israel.

The collaboration with the vaccination companies can be defined as a win-win situation without hesitation.

Getting ready for executing the mission – this paragraph should include a description of the relations existing in Israel between the ministry of health and the HMO's. And be unified with the paragraph of the centralized management of the health system that enabled to both prepare and execute the operation.

Maximal usage of the vaccine doses – there was indeed an ambition for a maximum use of every dose. Yet, there were several cases within which dozens of expensive doses were thrown. Specifying this will not lessen or damage the success of the vaccination operation.    

It is possible to enter a new sub-title in line no.241 since this paragraph relates to a principle indicating of community and media relations.

After action analytics – the content of this paragraph mixes between MCI(E) principles that should be mentioned in the introduction part of the article and a discussion of aspects that concerns after action analytics.

A summary which redirects readers to the main contention of the article as well as to future research plans, in or outside Israel, in relation to the vaccination operation in it or in other countries or in relation to MCE management principles in different disruption settings is missing.    

Author Response

Response to reviewer #1

The article analyzes the recent experience of the state of Israel in its COVID-19 vaccination operation. The authors contend that decision making processes during emergencies, such as COVID-19, should follow mass casualty event methodology.

The article is both interesting and important. It assists to provide initial understanding of the managing principles that helped Israel to successfully implement its vaccination operation during an emergency caused by an unfamiliar epidemic and in an uncertain, unpredictable context. Nevertheless, the article needs further clarifications and editing work. In the following paragraphs I will try to elaborate on my suggestions for editing the article.  

Answer: Thank you very much for the positive feedback, and useful comments and suggestions. We have carefully corrected and accepted your suggestions, and believe that the article has improved.

  • Introduction– the introduction lacks a funnel-like structure. After introducing the challenges posed in managing a pandemic crisis it needs to describe and typify the vaccination operation in Israel and then present the article's contention.

Answer: we changed and edited the structure of the introduction according to the funnel-like structure.  

Since the authors lean on a previous model of principles for mass casualty event (Ashkenazi et al., 2010), these principles should be presented clearly. Further then the authors should present the reasons this principal model needed translation and adaptations and specify their offered adapted principal model clearly – both in the text and through drawing a figure. The figure can take two versions – a "thin" one containing the main principles and a more elaborated one that will also present sub-principles of the model.   

Answer: we created a new section and discussed the Ashkenazi et al 2010 model carefully. We didn't add figure, but presented and explained all the model's principles.

Note that Ashkenazi et al. (2010) use the definition: mass casualty event in their article, and not incident, and specify 5 principles while the authors mention 4. Also, it is highly important to mention that decision making involves processes since it is through and upon these processes that decisions, strategies, practices etc. are being agreed upon and implemented.

Answer: thank you for this valuable comment. We referred to all the model's components. We also Added Padan 2017 to emphasize the process in decision making

The authors should provide and define what was the main reason(s) that stood behind the purpose of "opening the economy and keeping life routines" – to my opinion the purpose was to preserve Israel's functional continuity and to prevent further damage to the economy and wellbeing of Israelis which approximately 40% of them reported a mental deficiency of some sort caused by side effects of the pandemic (quarantine, social isolation, job losses etc.).  

Answer: we edited this section and added explanation in the beginning of the article. As other reviewer noted that an introduction about the population is unnecessary, we edited the paragraph and didn't refer to the population behavior.

Paragraphs (sub-titles) 2-4: For the paragraphs to be understandable to readers I strongly suggest presenting the central principle at the beginning of each paragraph and then exemplify it through using decision, practices etc. implemented in the Israel vaccination operation and through using concrete examples.

Answer: we added at the beginning of each paragraph the central principle.

In paragraph 2 pls. specify why, from a sociological point of view and in your opinion, an effective vaccine was the ultimate solution for containing the pandemic in Israel. I believe that a social explanation for this conclusion stands in affinity to other social factors influencing diverse steps, practices and procedures chosen by Israeli decision makers during the pandemic in Israel.

Answer: we added our opinion from a Sociological perspective (line 140)

The collaboration with the vaccination companies can be defined as a win-win situation without hesitation.

Answer: we edited this part.

Getting ready for executing the mission – this paragraph should include a description of the relations existing in Israel between the ministry of health and the HMO's. And be unified with the paragraph of the centralized management of the health system that enabled to both prepare and execute the operation.

Answer: We added an explanation of the relationship between the Ministry of Health and HMOs. We also changed the title: Getting ready for executing the mission with Purchasing the vaccines as it is more accurate and reflecting the content of the paragraph. 

Maximal usage of the vaccine doses – there was indeed an ambition for a maximum use of every dose. Yet, there were several cases within which dozens of expensive doses were thrown. Specifying this will not lessen or damage the success of the vaccination operation.    

Answer: "we lowered the tone" to make the loss of vaccines less 'dramatic'

It is possible to enter a new sub-title in line no.241 since this paragraph relates to a principle indicating of community and media relations.

Answer: we added a new subtitle 6. Community and Media Relations in line 281.

After action analytics – the content of this paragraph mixes between MCI(E) principles that should be mentioned in the introduction part of the article and a discussion of aspects that concerns after action analytics.

Answer: we appreciate this important comment. We edited the introduction, the model's section and the discussion.

A summary which redirects readers to the main contention of the article as well as to future research plans, in or outside Israel, in relation to the vaccination operation in it or in other countries or in relation to MCE management principles in different disruption settings is missing.

Answer: We added a short general summary

Reviewer 2 Report

The authors describe an important topic which is of great interest to the international community, as initiating and implementing an effective vaccination campaign is an important element in containing the COVID-19 pandemic. Furthermore, analyzing the Israeli vaccination campaign through the prism of MCI guidelines is interesting and worthy of an in-depth scrutiny.

Nonetheless, the manuscript requires improvement in the following key areas:

  1. The article focuses on the vaccination campaign but in several sections in the introduction, there is a mix-up between the management of the pandemic itself vs the vaccination campaign. The issue of pandemic management is not relevant to this article and does not contribute to the understanding of the reasons that led to the successful management of the vaccination process.
  2. The authors should support their varied statements by citing the references. Elements such as stated in lines 47-50 (similar situations in the different countries, availability of vaccines etc) must be supported by valid sources.
  3. It is unclear from what the authors stated, how the "scoop and run" policy that is characteristic to some systems in MCIs is applicable to the vaccination campaign (lines 119-120). Such a statement should be well explained, and supported by data.
  4. The authors present a high level of trust in the authorities and in the vaccines. There were actually low levels of trust in authorities during this stage of the COVID-19 pandemic. the authors should not disregard this, bring data from previous publications, and explain how this was overcome.
  5. The uniqueness of the HMO systems, which has a very high contribution to the success of the vaccination campaign should be further elaborated on. This is very different from what is used in MCI protocols, but proved in this campaign to be crucial.
  6. English editing is needed. Phrases such as "willingness to vaccine as much citizens....." are unacceptable, as well as the confusion between "doze" rather than "dose", are just two examples.

Author Response

Response to reviewer #2

Answer: Thank you very much for the valuable comments and suggestions. We corrected all needed, and appreciate their contribution to the quality and coherence of the manuscript.

The article focuses on the vaccination campaign but in several sections in the introduction, there is a mix-up between the management of the pandemic itself vs the vaccination campaign. The issue of pandemic management is not relevant to this article and does not contribute to the understanding of the reasons that led to the successful management of the vaccination process.

Answer: We deleted parts and edited the section dealing with the pandemic description and management

  1. The authors should support their varied statements by citing the references. Elements such as stated in lines 47-50 (similar situations in the different countries, availability of vaccines etc) must be supported by valid sources.

Answer: We rewrote this paragraph and the focus now is on Israel only.

2. It is unclear from what the authors stated, how the "scoop and run" policy that is characteristic to some systems in MCIs is applicable to the vaccination campaign (lines 119-120). Such a statement should be well explained, and supported by data.

Answer: We deleted this part, as we referred to the preparedness phase earlier in the manuscript. We also added a new comprehensive section on the MCE management model

3. The authors present a high level of trust in the authorities and in the vaccines. There were actually low levels of trust in authorities during this stage of the COVID-19 pandemic. the authors should not disregard this, bring data from previous publications, and explain how this was overcome.

Answer: We thank the reviewer for this comment. We added surveys' finding and explained the gap between the distrust f government officials and healthcare systems people (line 294)

4. The uniqueness of the HMO systems, which has a very high contribution to the success of the vaccination campaign should be further elaborated on. This is very different from what is used in MCI protocols, but proved in this campaign to be crucial.

Answer: Additional information about the HMOs system was added (from line 182).

5. English editing is needed. Phrases such as "willingness to vaccine as much citizens....." are unacceptable, as well as the confusion between "doze" rather than "dose", are just two examples.

Answer: We reviewed the entire manuscript again. We thank the reviewer for noticing.

Reviewer 3 Report

This article is well written and interesting about how other countries could adapt their response to COVID19, therefore it is interesting to publish it.

Howewer i have a major concern, on 13 citations it seems like 10 of those are self citations, such as the sentence line 153 with 3 lines and 3 self citations for very old references. Enhanced bibliography must be done.

More over, figure 2 is cropped.

Also, the bibliography format is not uniform as it should be for MDPI publication, and there is a lot of web links which are not permalink and thus will be lost in a more or less far future.

Each statement done in part 2 of the paper should be linked to a publication to assess the affirmation, even from Israeli government in Hebrew to at least find the original content stated.

Author Response

Response to reviewer #3

This article is well written and interesting about how other countries could adapt their response to COVID19, therefore it is interesting to publish it.

Answer: Thank you very much for the valuable comments and suggestions. We corrected all needed, and appreciate their contribution to the quality and coherence of the manuscript.

Howewer i have a major concern, on 13 citations it seems like 10 of those are self citations, such as the sentence line 153 with 3 lines and 3 self citations for very old references. Enhanced bibliography must be done.

Answer: We appreciate this comment. we added more references along the manuscript. The three mentioned citations are the most relevant we found.

More over, figure 2 is cropped.

Answer: We fixed the figure.

Also, the bibliography format is not uniform as it should be for MDPI publication, and there is a lot of web links which are not permalink and thus will be lost in a more or less far future.

Answer: We rewrote the bibliography. The web links direct to official websites (of the Ministry of Health for example), so we hope it will not be lost. We converted some web links to references.

Each statement done in part 2 of the paper should be linked to a publication to assess the affirmation, even from Israeli government in Hebrew to at least find the original content stated.

Answer: Thank you for this comment. We added references to the sources as much as possible.

Reviewer 4 Report

very interesting article that meets the scope and the standards of the journal.

Clear structure, sound methodology, argumentation supported by recent bibliography.

very minor syntax and grammar errors , approptiate writting tone.

acceptable referencing system 

a short conclusion needs to be added 

Author Response

Response to Reviewer #4

very interesting article that meets the scope and the standards of the journal.

Clear structure, sound methodology, argumentation supported by recent bibliography.

very minor syntax and grammar errors , approptiate writting tone.

acceptable referencing system 

a short conclusion needs to be added

Answer: Thank you very much for for the positive feedback. We added a short conclusion at the end of the manuscript

Round 2

Reviewer 2 Report

The manuscript was modified as needed and is now worthy of publication

Reviewer 3 Report

NA